# THP-1 Monocytic Cells Are Polarized to More Antitumorigenic Macrophages by Serial Treatment with Phorbol-12-Myristate-13-Acetate and PD98059

**DOI:** 10.3390/medicina60061009

**Published:** 2024-06-20

**Authors:** Hantae Jo, Eun-Young Lee, Hyun Sang Cho, Md Abu Rayhan, Ahyoung Cho, Chang-Suk Chae, Hye Jin You

**Affiliations:** 1Cancer Microenvironment Branch, Division of Cancer Biology, Research Institute, National Cancer Center, Goyang 10408, Republic of Korea; jesuswh1@gmail.com (H.J.); eylee@ncc.re.kr (E.-Y.L.); 76030@ncc.re.kr (H.S.C.); 76961@ncc.re.kr (A.C.); csc2022@ncc.re.kr (C.-S.C.); 2Department of Cancer Biomedical Science, National Cancer Center-Graduate School of Cancer Science and Policy, National Cancer Center, Goyang 10408, Republic of Korea; 99065@ncc.re.kr

**Keywords:** THP-1, polarization modulation, MEK-ERK pathway, PD98059

## Abstract

*Background and Objectives*: As modulators of the tumor microenvironment, macrophages have been extensively studied for their potential in developing anticancer strategies, particularly in regulating macrophage polarization towards an antitumorigenic (M1) phenotype rather than a protumorigenic (M2) one in various experimental models. Here, we evaluated the effect of PD98059, a mitogen-activated protein kinase kinase MAPKK MEK1-linked pathway inhibitor, on the differentiation and polarization of THP-1 monocytes in response to phorbol-12-myristate-13-acetate (PMA) under various culture conditions for tumor microenvironmental application. *Materials and Methods*: Differentiation and polarization of THP-1 were analyzed by flow cytometry and RT-PCR. Polarized THP-1 subsets with different treatment were compared by motility, phagocytosis, and so on. *Results*: Clearly, PMA induced THP-1 differentiation occurs in adherent culture conditions more than nonadherent culture conditions by increasing CD11b expression up to 90%, which was not affected by PD98059 when cells were exposed to PMA first (post-PD) but inhibited when PD98059 was treated prior to PMA treatment (pre-PD). CD11b^high^ THP-1 cells treated with PMA and PMA-post-PD were categorized into M0 (HLA-DR^low^ and CD206^low^), M1 (HLA-DR^high^ and CD206^low^), and M2 (HLA-DR^low^ and CD206^high^), resulting in an increased population of M1 macrophages. The transcription levels of markers of macrophage differentiation and polarization confirmed the increased M1 polarization of THP-1 cells with post-PD treatment rather than with PMA-only treatment. The motility and cytotoxicity of THP-1 cells with post-PD treatment were higher than THP-1 cells with PMA, suggesting that post-PD treatment enhanced the anti-tumorigenicity of THP-1 cells. Confocal microscopy and flow cytometry showed the effect of post-PD treatment on phagocytosis by THP-1 cells. *Conclusions*: We have developed an experimental model of macrophage polarization with THP-1 cells which will be useful for further studies related to the tumor microenvironment.

## 1. Introduction

The tumor microenvironment (TME), comprised of various cell types and other components, has been recognized as a critical determinant of tumor growth and progression [1]. Macrophages, the innate immune cells that constitute the first line of defense against damage or infection, can be attracted to the TME by modulating their polarization [2,3]. Tumor-associated macrophages indeed play a pivotal role in modulating the TME, forming the basis of several therapeutic strategies against cancers [1,4,5,6,7,8,9]. Packing of interferon (IFN)-γ into macrophage-targeting particles conserved antitumor phenotypes, resulting in a reduced tumor metastatic burden and slower tumor growth in animal models [10].

Differentiated macrophages are categorized as M1 pro-inflammatory and M2 anti-inflammatory types, which can be distinguished based on their cytokine and surface protein expression profiles [11,12,13]. M1 macrophages produce CXCL10, IFN-γ, interleukin (IL)-1β, IL-6, IL-12, IL-23, tumor necrosis factor (TNF)-α, reactive oxygen species (ROS), and nitric oxide (NO). M2 macrophages produce IL-10, CD163, CD206, transforming growth factor (TGF)-β, transglutaminase 2 (TGM2), and macrophage colony-stimulating factor 1 (CSF1) [2,8,9,11,13,14,15]. Phorbol-12-myristate-13-acetate (PMA), a homologue of diacylglycerol (DAG), activates a variety of signaling pathways through binding its receptor proteins such as protein kinase Cs, α-/β-chimaerins, and Ras guanyl-releasing protein 1 (RasGRP1) [16,17,18] and is used to activate and induce differentiation of human monocytic THP-1 [19,20].

THP-1 cells were derived from a 1-year-old boy with acute monocytic leukemia [21], and have been used in a number of studies of macrophages [4,20,22,23,24] as well as acute myeloid leukemia [25]. As a model of monocyte/macrophage, THP-1 cells are useful for mimicking the TME in vitro. To investigate the functions of polarized macrophage subsets in the TME, THP-1 cells have been exposed to PMA or vitamin D3 [22,23,26] to induce their activation or differentiation; bacterial cell wall components (including lipopolysaccharide) and IFN-γ to induce their polarization into M1 pro-inflammatory macrophages; and PMA, IL-4, and/or IL-13 to induce their polarization into M2 anti-inflammatory macrophages [23,27]. Proteomics analysis of THP-1 cells exposed to a variety of stimuli implicated mitogen-activated protein kinase kinase (MAPKK, MEK1) in peroxisome proliferator-activated receptor (PPAR)-γ-induced retinoic acid signaling, which leads to M2 polarization [28], implying a possible way to regulate macrophage polarization to pro-inflammatory, after differentiation.

In this study, we examined the impact of PMA on macrophage activation-induced polarization in various culture conditions, and the subsequent effects of inhibitors, such as PD98059, targeting MEK1 signaling in THP-1 cells to replicate the TME in vitro utilizing cancer cells and appropriately polarized macrophages.

## 2. Materials and Methods

### 2.1. Materials

RPMI-1640 and DMEM were obtained from Cytiva (South Logan, UT, USA) and defined fetal bovine serum (FBS) was purchased from GIBCO (Grand Island, NY, USA). Fluorochrome-labelled antibodies against CD11b, HLA-DR, CD279, and CD206 for flow cytometry were purchased from BD pharmingen, or BD Horizon (BD Biosciences, San Jose, CA, USA). PD98059 (MEK1 inhibitor) was obtained from Calbiochem^®^ Inc. (Merck KGaA, Darmstadt, Germany). Trametinib (GSK1120212) was purchased from Selleck Chemicals LLC. (Houston, TX, USA). CellTraceTM Calcein red-orange AM was obtained from InvitrogenTM (Life Technologies, Thermo Fisher Scientific Inc., Carlsbad, CA, USA). Calcein AM and all other chemicals were obtained from Sigma-Aldrich (St. Louis, MO, USA) unless otherwise specified.

### 2.2. Cell Culture

Human monocyte cell line, THP-1 (40202), human gastric cancer cell lines, SNU668 (00668), and SNU216 (00216) were obtained from Korean cell bank (KCLB, Seoul, Republic of Korea). The human colon carcinoma cell line, HCT116, and human non-small cell lung cancer cell lines, A549 and H460, were obtained from the American Type Culture Collection (Manassa, VA, USA). KNCC-STS2 cell line was newly established at National Cancer Center, South Korea [29]. All cells were authenticated by short-tandem repeat (STR) polymerase chain reaction (PCR) method in 2022 and 2023 by NCC genomic core facility. HCT116 cells were maintained as monolayers in DMEM, supplemented with 10% heat-inactivated FBS, 1% antimycotic-antibiotic solution (Thermo Scientific, Waltham, MA, USA). A549, H460, SNU668, SNU216, and THP-1 cells were maintained in RPMI 1640, supplemented with 10% heat-inactivated FBS, 1% antimycotic-antibiotic solution (Thermo Scientific, Waltham, MA, USA). For THP-1 maintenance, 25 mM HEPES and 55 μM 2-mercaptoethanol (Gibco, Grand Island, NY, USA) were additionally supplemented in suspension culture. All cells were grown at 37 °C in a humidified 5% CO_2_ atmosphere.

For differentiation and polarization, THP-1 cells (5 × 10^5^ cells/mL) were cultured in 96-well plates (analysis of cell adherence, Appendix A), 60 mm plates (observation or staining), 75 T flasks (adherent condition), and 50 mL tubes (nonadherent condition) for flow cytometry, or RNA/protein expression, and treated with PMA or DMSO for the indicated times. A number of 50 mL tubes were utilized to establish a nonadherent culture condition for THP-1 cells during PMA-driven differentiation. For some experiments, THP-1 cells were exposed to PD98059 for 30 min prior to (pre-PD) or 24 h after (post-PD) PMA treatment. The conditioned medium (CM) from THP-1 cells treated with PMA in the presence or absence of PD98059 (CM1-PMA and CM2-PMA-post-PD) was collected and its effect on cancer cell survival (cytotoxicity) was investigated. Some cells were treated with Trametinib (10 nM) in the presence or absence of PMA treatment for comparison.

### 2.3. THP-1 Differentiation and Polarization Analysis by Flow Cytometry

Adherent and nonadherent cells were harvested using Accutase (Innovative Cell Technologies, San Diego, CA, USA) and resuspended in FACS buffer [1× phosphate-buffered saline (PBS) with 2% fetal bovine serum (FBS)]. Cells were reacted with fluorochrome-conjugated antibodies against CD11b, HLA-DR, CD206, and CD279 (Appendix A) for 30 min at 4 °C in the dark. Cells were washed twice with FACS buffer, centrifuged at 500× *g* for 5 min, and resuspended in 500 μL of FACS buffer for flow cytometry using a LSRFortessa (BD Biosciences, San Jose, CA, USA); the data were analyzed using FlowJo™ v9 software (BD Biosciences). For obtaining CD11b^high^ macrophages for further experiments such as RT-PCR, harvested cells were reacted with an antibody against human CD11b (BD Biosciences), and subjected to sorting using the FACS Melody (BD Biosciences).

### 2.4. RNA Extraction and Reverse Transcription (RT) and PCR

Total RNA was extracted with the RNeasy kit (Qiagen, Valencia, CA, USA) and quantified. RNA samples (5 µg) were reverse transcribed at 42 °C for 60 min in 50 µL buffer (10 mM Tris, pH 8.3, 50 mM KCl, 5 mM MgCl_2_, and 1 mM dNTP) in the presence of oligo dT primer. Hot-start PCR was performed to increase the specificity of amplification with gene-specific primers (~10 pmoles/reaction, Appendix A). The PCR products were subjected to electrophoresis on 1.5–2% (*w*/*v*) agarose gels, and the resulting bands were visualized with ethidium bromide and photographed using the GelDoc program (Azure 200, Dublin, CA, USA). PCR products were sequenced for validation by Sanger sequencing method of Macrogen Inc. (Seoul, Republic of Korea). Quantification of band intensity was measured with ImageJ program (v1.53, National Institutes of Health, Bethesda, MD, USA).

### 2.5. Immunoblotting

Equal amounts of proteins were mixed with 2 × sample buffer and heated at 95 °C for 7 min and separated by sodium dodecyl sulfate-polyacrylamide gel electrophoresis on 8–10% acrylamide gels followed by transfer to polyvinylidene difluoride membranes. Immunoblotting was performed as described previously [29].

### 2.6. Cell Morphology by Microscopy

Cells on plates or glass slides were digitized using various microscopy techniques to observe their morphological characteristics. For light microscopy, the cells were digitized using inverted light microscopy (CKX53, Olympus, Tokyo, Japan). For holotomography, THP-1 cells were cultured on a 50 mm dish (Tomocube, Daejeon, South Korea) for 24 h, followed by the described treatments. The cells were then directly observed using HT-X1 (Tomocube, Daejeon, Republic of Korea) [30]. Images were processed in TomoStudio^TM^ and ImageJ (v 1.53). Some cells on slides were fixed in 4% paraformaldehyde and permeabilized in 0.25% Triton X-100 in PBS at room temperature for 10 min. The cells were stained with rhodamine phalloidin for 1 h at room temperature to label filamentous actin (F-actin) [31]. For nuclear staining, DAPI was applied to the cells at the final step for 5–10 min, and these were mounted with VECTASHIELD**^®^** Antifade Mounting Medium (Vector Laboratories, Inc., Newark, CA, USA).

### 2.7. Migration Assay

THP-1 cells, treated with PMA or PMA-post-PD, were harvested and stained with 0.5 μM Calcein AM (Sigma-Aldrich) for 30 min at 37 °C. After washing, cells (3 × 10^5^) were placed in the upper compartment of an 8 μm Transwell insert (Corning, Kennebunk, ME, USA) [32]. The lower compartment was filled with DMEM supplemented with 10% FBS. In some experiments, cancer cells were seeded the day before adding THP-1 cells. After 9 h, the filter of the upper compartment was washed. The migrated THP-1 cells on the filter membrane were fixed in 4% formaldehyde for 10 min and removed by wiping with a cotton swab. Assays were conducted in at least triplicate, and five random fields at 20× magnification were visualized using the Axio Observer Z1 Fluorescence Microscope (Carl Zeiss AG, Oberkochen, Germany). The obtained images were analyzed to count cells in each field using Image J.

### 2.8. Phagocytosis Assay

HCT116 cells (5 × 10^5^ cells/well) were grown on cover glasses or six-well plates for 24 h. THP-1 cells with PMA, PMA-post-PD, or DMSO (vehicle) treatment were harvested using Accutase and stained with Calcein AM (0.5 μM) in medium for 30 min at 37 °C. After washing with PBS, cells were stained with 0.5 μM Calcein AM (THP-1 cells) or 5 μM Calcein red-orange AM (HCT116 cells) for 30 min. Stained THP-1 cells (5 × 10^5^) were placed onto HCT116 cells for 6 h at 37 °C and subjected to confocal microscopy or FACS analyses of double fluorescence-labelled cells (phagocytosis). For confocal microscopy, cells on cover glasses were fixed in 4% formaldehyde and mounted in mounting medium (VECTOR Laboratories, Burlingame, CA, USA). Cells were visualized under a confocal microscope (LSM 880 Airyscan, Zeiss, Oberkochen, Germany) [31]. For FACS analysis, cells were harvested using Accutase and resuspended in FACS buffer for flow cytometry on a BD FACSLyric (BD Biosciences, San Jose, CA, USA). Data were analyzed using FlowJo™ v9 software to estimate cell populations based on the results of staining with two fluorescent dyes.

### 2.9. Statistical Analysis

All data are expressed as percentages of the control and shown as means ± standard error (SE). Between-group comparisons were performed by two-way analysis of variance and *t*-test in Prism software (v. 9.0, GraphPad Software, La Jolla, CA, USA). Values of *p* < 0.05 were considered statistically significant.

## 3. Results

### 3.1. PD98059 Influences the Differentiation and Polarization of THP-1 Cells in Response to PMA

PMA induces significant changes in THP-1 cells. However, the experimental approaches vary in each study. THP-1 differentiation in response to PMA needs to be evaluated based on several aspects, including treatment time and culture conditions, to ensure reproducibility. Notably, cells transition from a nonadherent to an adherent state, which may serve as the initial indication of differentiation, is as described in our previous report [20]. Nevertheless, it remains unclear whether the differentiation in response to PMA is influenced by the culture conditions, such as nonadherent conditions in tubes or adherent conditions in dishes. In other words, we are investigating if differentiated cells in response to PMA need adherent culture conditions for further process after transition to an adherent state. Therefore, THP-1 cells were cultured in 50 mL tubes (no area for adherence) and T75 flasks in the presence of PMA or DMSO (vehicle) for 72 h and subjected to flow cytometry for the marker of differentiation CD11b (Figure 1A). With gating for viability, the number of CD11b^high^ cells (Figure 1B,C) was greater when cultured under adherent than nonadherent conditions. The number of viable cells in the nonadherent culture was <60% of that with DMSO and <40% of that with PMA, whereas the viable cell population in the adherent culture was 80–90% of that with DMSO and ~60% of that with PMA. Furthermore, PMA-induced differentiation was greater in the adherent culture (9.865% ± 1.440% DMSO vs. 92.46% ± 0.67% PMA) than the nonadherent culture (29.58% ± 1.59% DMSO vs. 86.16% ± 1.26% PMA) (Figure 1C). Some cultures were treated with PD98059, a MEK1 pathway inhibitor, for 30 min prior to PMA treatment (pre-PD) (Figure 1A) to investigate the involvement of the MEK1 pathway in PMA-induced differentiation of THP-1 cells (Figure 1). Pre-PD treatment reduced the CD11b^high^ THP-1 cell population from 92.46% to 22.43% in the adherent culture, thus, supporting the role of the MEK1 pathway in PMA-induced differentiation of THP-1 cells. Adherent culture conditions clearly enhanced THP-1 differentiation and survival in response to PMA. Therefore, all further experiments regarding differentiation and polarization were conducted under adherent culture conditions to ensure the efficiency and reproducibility of our results. The CD11b^high^ population was not markedly affected by PD98059 treatment when cells were first treated with PMA for 24 h followed by PD98059 (PMA-post-PD) (92.46% to 90.02%), suggesting that PMA-post-PD did not affect differentiation of THP-1 cells in response to PMA.

As differentiated THP-1 cells exhibited the expression and secretion of various cytokines associated with pro-inflammatory or anti-inflammatory responses, we investigated the presence of cell populations polarized for M1 and M2 phenotypes in PMA-stimulated THP-1 cells. In addition, we explored the potential involvement of MEK1-linked pathways in differentiated THP-1 cells, in response to PMA stimulation, as described previously [28]. For this purpose, THP-1 cells were labeled with PMA alone or with PMA followed by PD98059 (PMA-post-PD) using fluorochrome-labeled antibodies against CD11b, HLA-DR, CD206, or CD279 (PD-1) for subsequent analysis by flow cytometry. About 90% of CD11b^high^ cells were further gated for M0 (HLA-DR^low^ and CD206^low^), M1 (HLA-DR^high^ and CD206^low^), and M2 (HLA-DR^low^ and CD206^high^) (Figure 2A,F, Appendix A). Among THP-1 cells treated with PMA, 62.06% were M0, 34.61% were M1, and 1.02% were M2. Among PMA-post-PD THP-1 cells, 43.43% were M0, 55.30% were M1, and 0.22% were M2 (Figure 2B,C). Therefore, PMA likely induced THP-1 cell differentiation via the MEK1 pathway (pre-PD treatment, Figure 1C) and PMA-induced differentiated THP-1 cells may become markedly polarized to M1 in the presence of PD98059 (PMA-post-PD) (Figure 2B). Simultaneously, cells were observed by microscopy to optimize treatment conditions (Appendix A).

CD274 (PD-1) on macrophages has been implicated in phagocytosis [6] and is predictive of the clinical benefit of anti-PD-1 therapy [33]. We quantified THP-1 cells expressing PD-1 (Figure 2C,D). The PD-1^+^ cell population varied by approximately 1% among the subsets. Of the THP-1 cells treated with PMA, 1.52% were PD-1^+^ compared to 0.83% for PMA-post-PD cells; most of the reduction was in the M1 population (Figure 2D). These results suggest that the MEK1 pathway promotes phagocytosis by macrophages via modulation of the PD-1 expression. Next, we considered to get further insight into the clinical application of anticancer drugs on macrophage polarization by targeting MEK1 pathways. Trametinib, which has been approved for the treatment of some cancers with BRAF mutations by the United States Food and Drug Administration (FDA), targets MEK1 and MEK2 [34,35]. We further compared the effect of Trametinib and PD98059 on THP-1 differentiation and polarization in response to PMA by flow cytometry (Figure 2F). CD11b^high^ populations of THP-1 with PMA cells in the presence of Trametinib (10 nM) reached 80–90%, implying that PMA-post-PD98059 PMA-post-Trametinib showed similar effect on THP-1 differentiation in response to PMA. In addition, differentiated THP-1 cells showed a greater M1 population in the presence of Trametinib compared to PMA alone, as expected, implying the role of the MEK-linked pathway in the polarization of differentiated THP-1 cells in response to PMA. Interestingly, PD98059 demonstrated a larger M1 population than Trametinib in differentiated THP-1 cells, supporting the idea that MEK1 and MEK2 might contribute differentially to polarization.

### 3.2. PD98059 Affects the Expression Levels of Genes Related to Macrophage Differentiation and Polarization

Based on their expression of surface markers, THP-1 cells were differentiated (~90%) and polarized to M1 (~35%) in response to PMA, which was enhanced (~55%, 1.57-fold more) by PMA-post-PD treatment. We next examined the expression of genes related to macrophage differentiation and polarization in THP-1 cells treated with PMA, PMA-post-PD, or DMSO (vehicle) (Figure 3 and Appendix A). PD98059 influenced the expression of genes related to polarization (Figure 3A). The expression levels of genes related to M1 polarization, iNOS, IFN-γ, CXCL10, and TNF-α were increased by PMA-post-PD treatment in PMA-stimulated THP-1 cells, whereas those of genes related to M2, IL-10, CD206, and TGM2 were decreased (Figure 3A). The expression of CD68 [36] was decreased by PMA-post-PD treatment in THP-1 cells, suggesting that it is more than merely a marker of macrophages [37]. In addition, we sorted CD11b^high^ THP-1 cells treated with DMSO, PMA, or PMA-post-PD and examined their expression levels of several genes by RT-PCR (Figure 3B). We were unable to obtain sufficient number of DMSO-treated cells, so these were excluded from the analysis. In CD11b^high^ cells, PD98059 treatment increased the expression of CXCL10 and decreased those of CD206, CSF1, and TGM2, implying an altered M1 population. The expression levels of M2 markers were modulated by PD98059, although this cell population comprised <1% of the total. As suggested by He et al. [28], differentiated THP-1 might be polarized by more anti-inflammatory M1 by targeting the MEK1-ERK pathway by PMA-post-PD treatment, which implies that cytokines and receptors such as IL-4 and IL-4R from differentiated THP-1 might activate the MEK-ERK pathway. Immunoblotting analysis demonstrated the effects of PD98059 on ERK phosphorylation in PMA-differentiated THP-1 cells (Figure 3C,D). As a direct target of MEK1/2, ERK phosphorylation was affected by PD98059 treatment in PMA-stimulated THP-1 cells.

### 3.3. PD98059 Increases the Motility of Differentiated THP-1 Cells

Macrophages move from the peripheral blood or tissues to sites of differentiation by cytokines or pathogen-derived factors, such as lipopolysaccharide (LPS) [9]. On holotomography (Figure 4A) [30], THP-1 cells treated with PMA or PMA-post-PD showed increased adherence to plates with a level of focal adhesion comparable to those treated with DMSO (vehicle), implying actin filament reorganization. PMA-post-PD treatment increased the number of focal adhesions (Figure 4A). Functionally, the number of migrating THP-1 cells was increased more than two-fold by PMA-post-PD compared to PMA treatment in a Transwell assay (Figure 4B,C). The motility of THP-1 cells with PMA-post-PD treatment was also enhanced when cancer cells were placed in the bottom chamber of the Transwell system (Figure 4D). THP-1 cells with PMA-post-PD treatment exhibited 4–13-fold greater migration toward HCT116, A549, H460, SNU668, and KNCC-STS2 sarcoma cells [29], as compared to that in the absence of cancer cells. These results suggest that cell migration is enhanced by communication between differentiated macrophages and cancer cells in a cancer cell type- or context-specific manner.

### 3.4. PD98059 Promotes Phagocytosis by Differentiated THP-1 Cells

As innate immune cells, macrophages phagocytose bacteria, viruses, dying cells, and cancer cells [7]. We investigated further whether phagocytic activity increased with the increase in the M1 population among THP-1 cells with PMA-post-PD treatment. We labeled THP-1 cells with Calcein AM and HCT116 cells with Calcein red-orange AM for 30 min. Calcein red-orange AM-labeled THP-1 cells were added to HCT116 cells for 6 h; they were subsequently visualized by confocal microcopy and quantified by flow cytometry (Figure 5). Confocal microscopy in three-dimensional (3D) ortho view with ZEN 3.1 (Blue Edition) software (Carl Zeiss, Oberkochen, Germany) showed that some THP-1 cells with PMA or PMA-post-PD treatment had a Calcein AM signal with one or two nuclei (Figure 5A, yellow arrowhead, left and center panels). Some cells were subjected to 3D volume reconstruction (Figure 5A, right panel). For quantitative analysis, we again conducted flow cytometry (Figure 5B). The numbers of THP-1 cells with Calcein and Calcein red-orange fluorescence signals were greater with PMA-post-PD than PMA treatment (Figure 5C,D). Macrophages engaged in phagocytosis, which are larger than their nonphagocytosing counterparts, can be gated by a narrow singlet. We analyzed the large-cell population by gating for FSC-H and FSC-W (multi) and regular singlet gating (single) (Figure 5B). Both gating strategies showed a significantly increased population of double-positive THP-1 cells after PMA-post-PD treatment. The increase in the M1 population of THP-1 cells caused by PD98059 suggested enhanced phagocytosis of cancer cells, suggesting that PD98059 exerts an antitumor effect on PMA-treated THP-1 cells.

## 4. Discussion

The major findings of this study were that adherent THP-1 cells underwent more rapid and efficient differentiation than nonadherent cells (Figure 1). In the absence of other stimuli, THP-1 cells with PMA treatment had a 40% rate of M1 polarization (Figure 2), which was enhanced by subsequent treatment with PD98059 (Figure 2 and Figure 3). PMA-post-PD treatment increased the M1 population of THP-1 cells by up to 60% (up to 1.5-fold) and enhanced their motility and phagocytotic activity compared to those treated with PMA alone (Figure 4 and Figure 5). PMA-stimulated THP-1 cells underwent differentiation and polarization in both the absence and presence of LPS stimulation. Therefore, the cytokines and other factors induced by PMA, may be sufficient to induce the polarization of THP-1 cells.

After the onset of differentiation, the MEK1/ERK pathway may contribute to the M1 polarization driven by IFN-γ and CXCL10 (Figure 3). About 1% of THP-1 cells with PMA or PMA-post-PD treatment underwent polarization to M2, which showed the clear effects of PD98059 treatment (Figure 2 and Figure 3), strengthening the role of the MEK1/ERK pathway in M2 polarization, as reported previously [28].

The targeted therapeutics Mekinist^®^ (Trametinib) and/or Tafinar^®^ (Dabrafenib) have been approved for the treatment of some cancers with BRAF mutations by the United States FDA [34,35]. Trametinib, the agent targeting MEK1/2, could modulate macrophage populations, resulting in increased M1 population (Figure 2F). Thus, the benefit of targeting MEK1/2 for cancer cells with BRAF mutations and macrophage polarization by MEK1/2 inhibitors requires further investigation.

Interestingly, there are controversies in regard to the MEK-ERK pathway on macrophage polarization. Manicone and colleagues showed the MEK1/2 inhibitors resulted in greater M2 polarization in alveolar macrophages and bone marrow-derived macrophages in the presence of IL-4/IL-13 [38], which was further supported by another study to show MEK1/2 inhibition on pro-inflammatory response [39]. In contrast, MEK-ERK driven M2 polarization was selectively blocked by MEK and histone deacetylase inhibitors in THP-1 and macrophages prepared from mice and humans [28]. In this study, we have demonstrated that the MEK1-ERK pathway serves as a target to modulate the M1 population of THP-1 cells under PMA treatment. Controversy might be caused by a variety of experimental systems and types, such as treatment condition and processes, for evaluation and so on.

The limitation of this study might be lack of investigation into the effect of nutrients on THP-1 differentiation and polarization in response to PMA in the presence or absence of PD98059 or Trametinib. In the tumor microenvironment, nutrients, or metabolic environment, might be closely related to the MEK-ERK pathways and the efficacy of targeting these pathways. Further evaluation in other macrophages is warranted to assess responses to a variety of stimuli and nutrients.

Taken together, our findings suggested a role of the MEK1/ERK pathway in M1 macrophage polarization via multiple mechanisms in PMA-treated THP-1 cells. In addition, PD98059 inhibited M2 and enhanced M1 polarization from the M0 state. These findings contribute to our understanding of the crosstalk between tumors and macrophages in cell-based systems, paving the way for future research in this field.

## 5. Conclusions

In this study, we have shown that PMA-stimulated THP-1 cells express cytokines and factors for further polarization and that modulating the MEK-ERK pathway by PD98059 or Trametinib facilitates M1 polarized subsets, which will be useful for further studies related to the tumor microenvironment.

## Figures and Tables

**Figure 1 medicina-60-01009-f001:**
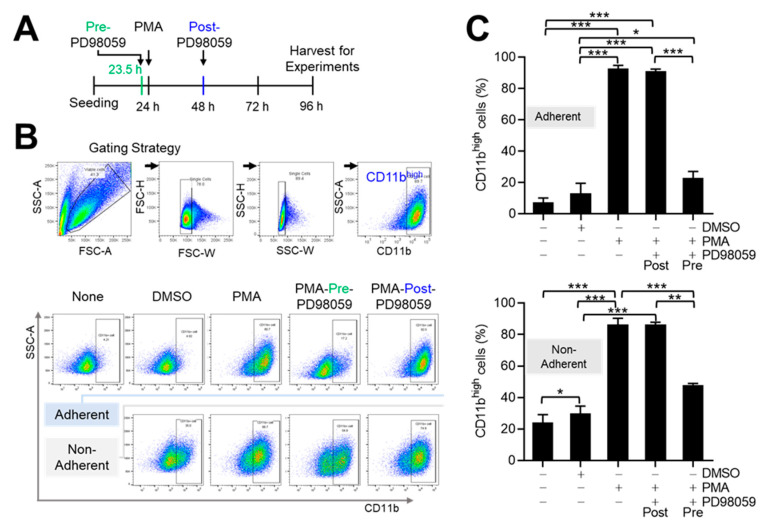
Effects of PD98059 on THP-1 cell differentiation in response to PMA. (**A**) Simplified experimental design for THP-1 differentiation and polarization by PMA and PD98059. (**B**) Gating strategy and dot plots for flow cytometry to obtain viable single THP-1 cells and to quantify CD11b^high^ populations. Dot plots of adherent and nonadherent CD11b^high^ THP-1 cells. Results are representative of at least four independent experiments. (**C**) Relative adherent and nonadherent populations of CD11b^high^ THP-1 cells. Data are means ± SE of at least four independent experiments (* *p* < 0.05, ** *p* < 0.001, and *** *p* < 0.0001).

**Figure 2 medicina-60-01009-f002:**
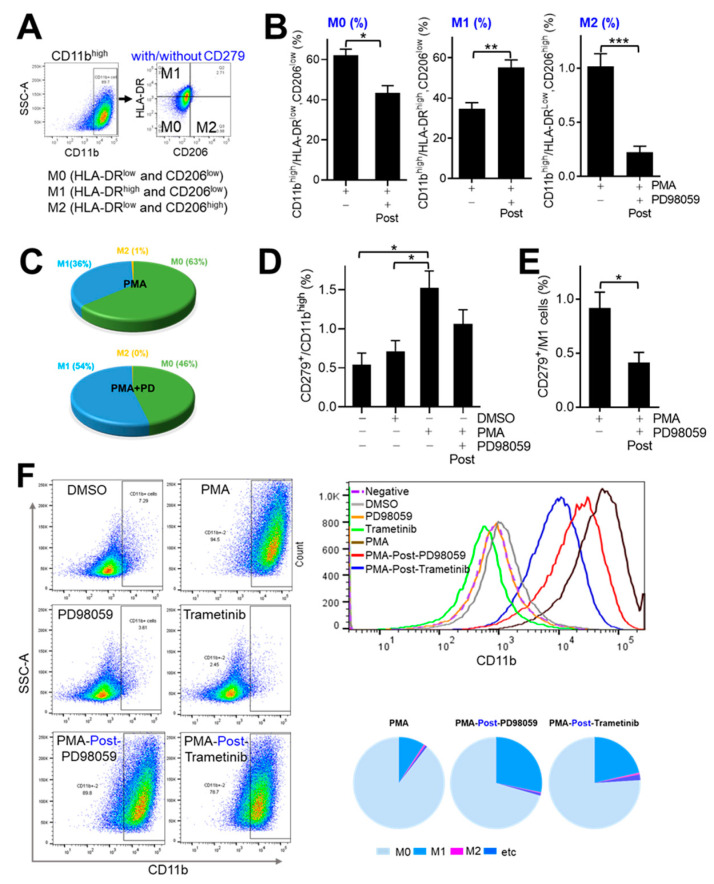
Effects of subsequent treatment with PD98059 or Trametinib on polarization of PMA-stimulated THP-1 cells by HLA-DR, CD206, and CD279. (**A**) Gating strategy for flow cytometry to quantify M1, M2, and M0 populations based on CD11b, HLA-DR, and CD206 expression. (**B**–**F**) Gated CD11b^high^ THP-1 cells from adherent cultures were gated for CD11b, HLA-DR, CD206, and CD279. (**B**) The M0, M1, and M2 populations of PMA-stimulated THP-1 cells according to subsequent 10 μM PD98059 (**B**–**E**) or 10 nM Trametinib (**F**) treatment. Some cells were treated with DMSO, PD98059, or Trametinib without PMA for comparison (**F**). Data are means ± SE of at least eight independent experiments (* *p* < 0.05, ** *p* < 0.001, and *** *p* < 0.0001). (**C**) Graphs of macrophage subsets in THP-1 cells treated with PMA with or without subsequent PD98059 treatment. (**D**) CD279 (PD-1)^+^/CD11b^high^ THP-1 cell populations. (**E**) CD279 (PD-1)^+^ M1 population of PMA-stimulated THP-1 cells with or without subsequent PD98059 treatment. (**F**) Dot plots of CD11b^high^ population of THP-1 cells in each treatment (left panel), histogram of CD11b^high^ specific population in each treatment (right top), and relative population of polarized THP-1 cells in each treatment (right bottom). Data shown are representative of two independent experiments.

**Figure 3 medicina-60-01009-f003:**
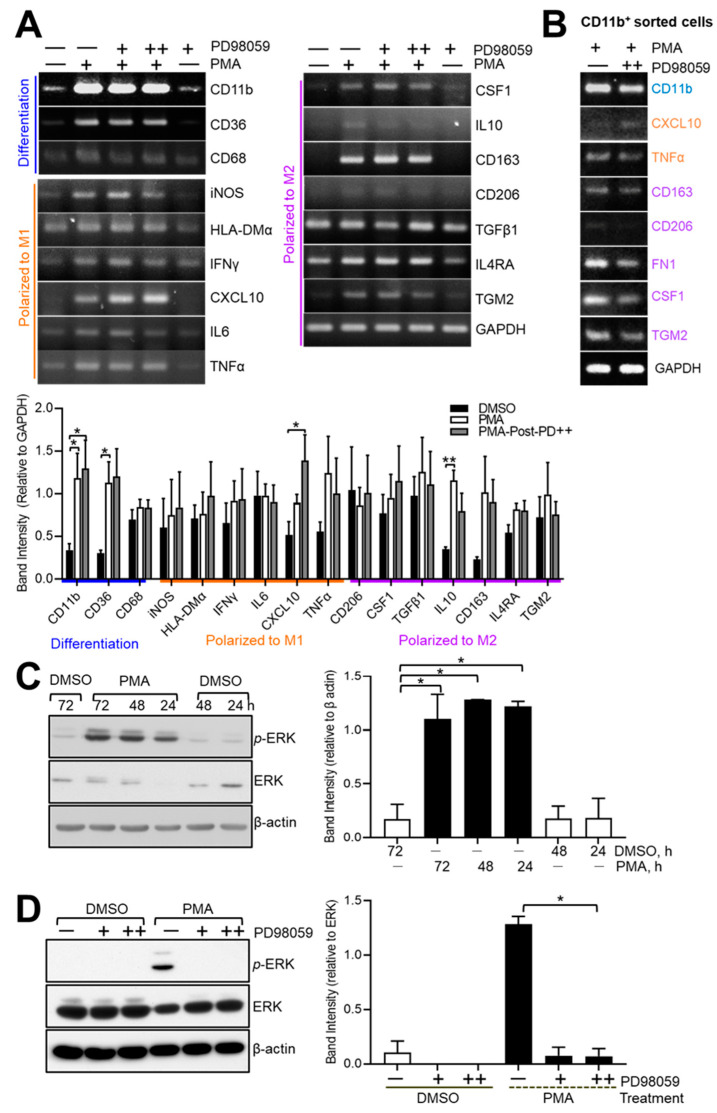
Effects of subsequent treatment with PD98059 on expression levels of differentiation- and polarization-related genes in PMA-treated THP-1 cells. (**A**,**D**) THP-1 cells were treated with 100 nM PMA or DMSO (vehicle) for 72 h and harvested for RT-PCR (**A**) or immunoblotting (**C**,**D**). Some cells were treated with 5 or 10 μM PD98059 for 48 h in the presence or absence of PMA. Some cells were subjected to flow cytometry to obtain CD11b^high^ cells (**B**) for RT-PCR. The band intensity was normalized by GAPDH and expressed as means ± SE of at least three independent experiments (* *p* < 0.05). (**C**) ERK1/2, downstream targets of MEK1, were activated in response to PMA in a time-course manner in THP-1 cells. Immunoblotting was performed with anti-ERK1/2, anti-phosphorylated ERK1/2, and β-actin (loading control). The band intensity was normalized by β-actin and expressed as means ± SE of at least three independent experiments (* *p* < 0.05). (**D**) Phosphorylation ERK1/2 were inhibited by PD98059, a MEK1 inhibitor, in PMA-stimulated THP-1 cells. The band intensity was normalized by β-actin and expressed as means ± SE of at least three independent experiments (* *p* < 0.05). Immunoblotting was performed with anti-ERK1/2, anti-phosphorylated ERK1/2, and β-actin (loading control). +, 100 nM PMA; +, ++: 5, 10 μM PD98059, respectively. Data are representative of at least three (**A**,**C**,**D**) and two (**B**) independent experiments.

**Figure 4 medicina-60-01009-f004:**
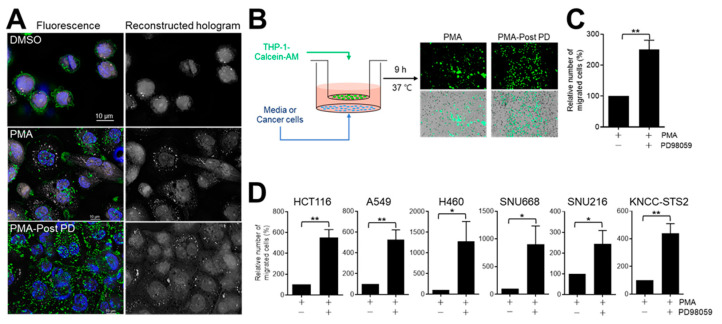
Effects of subsequent treatment with PD98059 on the motility of differentiated THP-1 cells. (**A**) THP-1 cells with PMA, PMA-post-PD, or DMSO treatment were stained with rhodamine phalloidin (F-actin) and Hoechst (nucleus), followed by holotomography (HT-X1; Tomocube, Daejeon, South Korea). Holograms were reconstructed using HT-X1 and ImageJ (v 1.53) software. Scale bar, 10 μm. (**B**) Cell migration assay using the Transwell system. (**C**,**D**) THP-1 cells (3 × 10^5^) with PMA or PMA-post-PD treatment were placed in the upper chamber of the Transwell system in the absence (**C**) or presence (**D**) of cancer cells for 9 h. Cancer cells were cultured in the bottom chamber of Transwell plates for 24 h before adding THP-1 cells. Migrated THP-1 cells with Calcein AM signals were imaged and enumerated using ImageJ. Data are means ± SE of at least four independent experiments (* *p* < 0.05 and ** *p* < 0.001).

**Figure 5 medicina-60-01009-f005:**
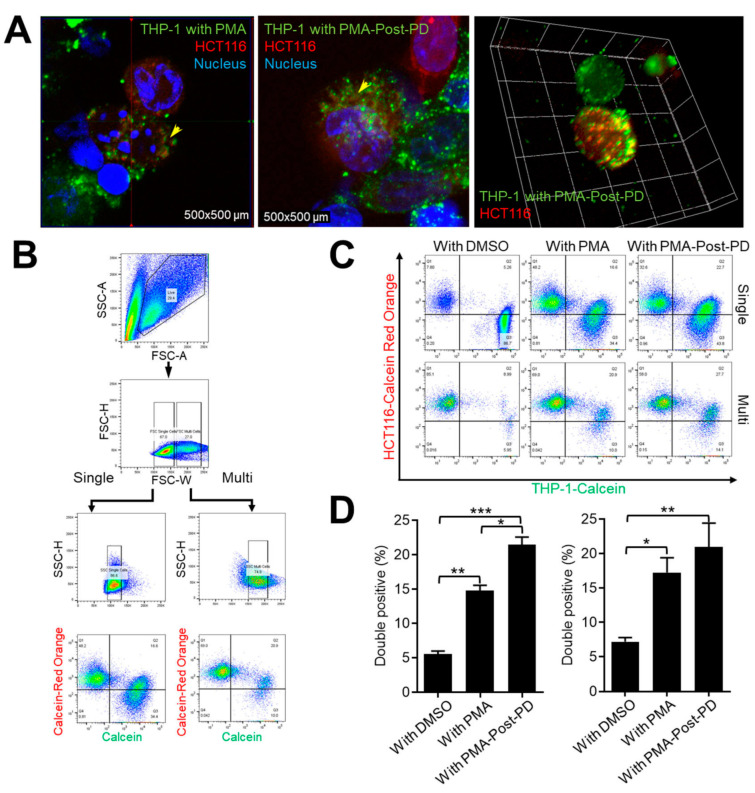
Effects of subsequent treatment with PD98059 on the phagocytic activity of differentiated THP-1 cells on HCT116 cancer cells. (**A**–**D**) HCT116 cells were cultured for 24 h on slide glasses (**A**) or six-well plates (**B**–**D**) and stained with Calcein red-orange AM for 30 min. THP-1 cells with PMA or PMA-post-PD treatment were stained with Calcein AM for 30 min and transferred to plates for analysis of phagocytosis for 6 h. (A) Fixed cells were observed by confocal microscopy (LSM 880 Airyscan; Carl Zeiss, Oberkochen, Germany), and 3D ortho views and 3D volume reconstructions were created using ZEN v. 3.1 (Blue Edition; Carl Zeiss) and ImageJ (v 1.53) software. (**B**) Gating strategy for quantifying phagocytosis by flow cytometry. (**C**,**D**) For quantification of phagocytosis, cells with Calcein (FITC-green) and Calcein Red-orange (PE-red) signals were analyzed. Representative dot blots of at least four independent experiments are shown (**C**) and data in (**D**) are means ± SE of at least four independent experiments (* *p* < 0.05, ** *p* < 0.001, and *** *p* < 0.0001).

## Data Availability

The data that support the findings of this work are available from the corresponding author upon request.

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
