# Peer review of "THP-1 Monocytic Cells Are Polarized to More Antitumorigenic Macrophages by Serial Treatment with Phorbol-12-Myristate-13-Acetate and PD98059"

_medicina, 2024, doi:10.3390/medicina60061009_

Round 1

Reviewer 1 Report

Comments and Suggestions for Authors

Review for JO Hanate et.al. “THP-1 monocytic cells might be polarized to more antitumorigenic macrophages with serial treatment of phorbol-12-myristate-13-acetate and PD98059”

General Comment:

This study with the title “THP-1 monocytic cells might be polarized to more antitumorigenic macrophages with the serial treatment of phorbol-12-myristate-13-acetate and PD98059” is suitable and fits within the scope of Medicine journal.

Specific Comments

Title:

1.       Remove “Maybe” from the title; simplify the title according to your conclusion.

Abstract:

1.       If possible, please provide a graphical abstract figure for readership.

Introduction:

1.       Provide detail on “How Protein Kinase C” and MAPK pathway are interlinked in the Introduction for a general audience. Line “56” Provides the details on some of those pathways; just stating the variety of pathways is incomplete and confusing for a general audience.

2.       Line “59” “Although there is some debate regarding their utility.” Provide a few examples for the same.

Results:

1.       Lines 197-198 provide a reference for adherence vs non-adherence culture and briefly describe the difference between both conditions.

2.       Restructure Figure 1, as Figures 1, B, and C cannot be cited before Figure 1A as per general convention, or revise your sentence.

3.       Panels B and C of Figure 1 are part of the same experiment, so combine both gating strategies followed by comparison.

4.       FACS X-axis and Y-axis labels are not visible. Simplify and revise them, for e.g., “use a bigger font for SSC-A, FSC-A; remove “CompPEcy7-CD11b” and simply write CD11B in bigger visible fonts.

5.       Figure 1C, What are CD11b negative cells? It might be CD11B low and CD11B high. Otherwise provide references for defining CD11B negative THP-1 cells.

6.       What is the logic for Adherent vs Non-Adherent THP-1 cell culture, and why is it important?

7.       Figure 2A, there is no clear separation for CD206 high population. How the gating has been decided. It seems the M2 population is just by error. Provide FMO or other control for deciding the gating line in supplemental.

8.       Simplify Figure Legend 2 and provide the Antibodies detail in a separate supplemental list.

9.       I could not understand why Trametinib is used and what the question is for using Trametinib. Describe in detail.

10.   Figure 3A, B, C: Did you perform any statistics for RT-PCR quantification? Showing Images is just qualitative data. It must be supplemented with quantification and statistics.

11.   Figure 3C, D. Is it only one n number?

12.   Line 287-287, either provide the data for the claim or remove this sentence from the manuscript.

Discussion:

1.       Overall, the discussion section is very poorly written. This section needs to provide a rich discussion comparing current findings and their importance, as well as citing more literature for comparison and possible mechanisms. The discussion should include a paragraph of limitation of the study.

2.       Line 386, either show the data or remove this claim from the manuscript.

3.       Lines 396-398 are confusing, and need to be revised.

Minor Corrections:

1.       Line 104, 173 correct “5 X 105” Cells

2.       Line 119-120 “ No antibodies are Fluorescein conjugated “ in the provided Supplemental table. The same mistake was repeated in the Results section, Line 233-234. If the Author used AI for editing language, please read carefully and disclose the usage of AI in decleration.

3.       Provide the version of the software used, like Flowjo , TomoStudio X and ImageJ.

4.       Line 137 says PCR products were sequenced. Where is the data for that sequencing? Which platform is used for sequencing?

5.       Figure 2E, labeling for PD98059 is incorrect.

6.       Line 266, correct Figure 1D to 2D.

Comments on the Quality of English Language

Author Response

NATIONAL CANCER CENTER

Cancer Microenvironment Branch, Division of Cancer Biology, Research Institute; Department of Cancer Biomedical Science, NCC-Graduate School of Cancer Science and Policy
323 Ilsan-ro, Ilsandong-gu, Goyang-si, Gyeonggi-do, 10408, Republic of Korea
Phone: +82-31-920-2206, Fax: +82-31-920-2006

May 24th, 2024

Professor, Dr. Edgaras Stankevicius Editor in Chief

Medicina, 

Editorial Office, MDPI Publications.

Dear Professor, Dr. Edgaras Stankevicius Editor in Chief,  

Thank you very much for the valuable comments on our manuscript, medicina-3005169, entitled “THP-1 Monocytic Cells May be Polarized to More Antitumorigenic Macrophages by Serial Treatment With Phorbol-12-Myristate-13-Acetate and PD98059”. 

Based on the reviewer’s comments, we are providing a point-by-point discussion of the points raised from the reviewers.

Reviewer 1:

General Comment:

This study with the title “THP-1 monocytic cells might be polarized to more antitumorigenic macrophages with the serial treatment of phorbol-12-myristate-13-acetate and PD98059” is suitable and fits within the scope of Medicina journal.

Specific Comments

Title:

  1. Remove “Maybe” from the title; simplify the title according to your conclusion.

As pointed by the reviewer, we edited the title from “THP-1 Monocytic Cells May be Polarized to ~” to “THP-1 Monocytic Cells Are Polarized to ~” in the revised manuscript.

Abstract:

  1. If possible, please provide a graphical abstract figure for readership.

As suggested by the reviewer, we tried to summarize our finding as a graphical abstract as shown below.

Introduction:

  1. Provide detail on “How Protein Kinase C” and MAPK pathway are interlinked in the Introduction for a general audience. Line “56” Provides the details on some of those pathways; just stating the variety of pathways is incomplete and confusing for a general audience.

As pointed by the reviewer, we improved the explanation as follows in the revised manuscript.

Line 53-57

Phorbol-12-myristate-13-acetate (PMA), a homologue of diacylglycerol (DAG), activates a variety of signaling pathways through binding its receptor proteins such as protein kinase Cs, α-/β-chimaerins and Ras guanyl-releasing protein 1 (RasGRP1) [16-18] and is used to activate and induce differentiation of human monocytic THP-1[19,20].

  1. Line “59” “Although there is some debate regarding their utility.” Provide a few examples for the same.

We are thankful for the valuable comments.  As suggested by reviewer, we edited the expression as follows to avoid any misunderstanding in the revised manuscript.  And one more reference “Reference No. 25” was added to show an example of THP-1 cell in acute myeloid leukemia study. 

THP-1 cells derived from a 1-year-old boy with acute monocytic leukemia [21], and have been used in a number of studies of macrophages [4,20,22-24] as well as acute myeloid leukemia [25].  As a model of monocyte/macrophage, THP-1 cells are useful for mimicking the TME in vitro.

Reference No #25 James, J.R.; Curd, J.; Ashworth, J.C.; Abuhantash, M.; Grundy, M.; Seedhouse, C.H.; Arkill, K.P.; Wright, A.J.; Merry, C.L.R.; Thompson, A. Hydrogel-Based Pre-Clinical Evaluation of Repurposed FDA-Approved Drugs for AML. Int J Mol Sci 2023, 24, doi:10.3390/ijms24044235.

Results:

  1. Lines 197-198 provide a reference for adherence vs non-adherence culture and briefly describe the difference between both conditions.

As pointed by the reviewer, we briefly described the culture condition in regard to adherence as follows and also added information in line 106-107 in materials and methods in the revised manuscript.

Line 200-203

In other words, it is investigated if differentiated cells in response to PMA need adherent culture condition for further process after transition to adherent state.  Therefore, THP-1 cells were cultured in 50-mL tubes (no area for adherence) and….

  1. Restructure Figure 1, as Figures 1, B, and C cannot be cited before Figure 1A as per general convention, or revise your sentence.

As pointed by the reviewer, we cited Figure 1A before Figure 1B and C as shown below, which was also revised in the revised manuscript.

…Therefore, THP-1 cells were cultured in 15-mL tubes and T75 flasks in the presence of PMA or DMSO (vehicle) for 72 h and subjected to flow cytometry for the marker of differ-entiation CD11b (Fig. 1A).  With gating for viability, the number of CD11b+ cells (Fig. 1B and C) was greater when cultured under adherent than nonadherent conditions.

  1. Panels B and C of Figure 1 are part of the same experiment, so combine both gating strategies followed by comparison.

As suggested by the reviewer, we combined Figure 1B and C for clarity.  Thank you for the opinion.

  1. FACS X-axis and Y-axis labels are not visible. Simplify and revise them, for e.g., “use a bigger font for SSC-A, FSC-A; remove “CompPEcy7-CD11b” and simply write CD11B in bigger visible fonts.

As pointed by the reviewer, we improved the labels through figures with FACS experiments in the revised manuscript.  Thank you for your valuable comments.

  1. Figure 1C, What are CD11b negative cells? It might be CD11B low and CD11B high. Otherwise provide references for defining CD11B negative THP-1 cells.

As pointed by the reviewer, we replaced CD11b+ cells to CD11bhigh cells for precise expression in the revised manuscript.  

  1. What is the logic for Adherent vs Non-Adherent THP-1 cell culture, and why is it important?

As studied by a lot of researchers, THP-1 cells are very useful model for monocytes/macrophages and convenient to deal with.  However, the experimental approaches are descriptive and had better be evaluated for clarity.  In this study, we tested several aspects of THP-1 differentiation and polarization in response to PMA from culture conditions to treatment time, and so on. Clearly, we showed THP-1 needs adherent condition for efficient differentiation and survival in response to PMA by showing cells in 50-ml tubes with less differentiation and poor viability during treatment time (Fig. 1).

  1. Figure 2A, there is no clear separation for CD206 high population. How the gating has been decided. It seems the M2 population is just by error. Provide FMO or other control for deciding the gating line in supplemental.

As suggested by the reviewer, we showed the control as figure below.  THP1 cells were exposed to PMA and harvested for FACS analyses.  Some cells were not stained at all and used as negative control of each antibody signal.  The other cells were stained with fluorochrome-conjugated antibodies against CD11b, CD206, HLA-DRα, and PD-1.  As shown in figure below, each antibody clearly showed antibody-specific fluorochrome signal. The signal from CD206 was relatively weaker than other markers however, clearly existed.  Thus our gating strategy was applied in this study.  

  1. Simplify Figure Legend 2 and provide the Antibodies detail in a separate supplemental list.

As pointed by the reviewer, we listed antibodies for this gating as shown below and added as a supplementary table for better understanding and tried to simplify the figure legend in the revised manuscript.

  1. I could not understand why Trametinib is used and what the question is for using Trametinib. Describe in detail.

In this study, we have demonstrated the involvement of MEK-ERK pathway on altered populations by PD98059 (specific for MEK1) in PMA-stimulated THP-1 cells in terms of polarized subsets.  Trametinib in PMA-stimulated THP-1 cells, showed increased M1 population in PMA-stimulated THP-1 cells, supporting the idea, MEK/ERK might be important for macrophage polarization to anti-cancer, and proinflammatory subset not only for differentiation.

By showing effect of Trametinib, our results were further confirmed.  In addition, the results from trametinib-experiments, are meaningful to provide the evidence for future direction in the therapeutic strategy against cancer with RTK, Ras, Raf alteration and certain immune microenvironment.   

  1. Figure 3A, B, C: Did you perform any statistics for RT-PCR quantification? Showing Images is just qualitative data. It must be supplemented with quantification and statistics.

As pointed by the reviewer, we quantified RT-PCR and immunoblotting results (Figure A, 5 repeats; Figure C, 3 repeats, and Figure D, 3 repeats, respectively) and statistically analyzed by GraphPad Ver.9. (Prizm), as shown in figure left.  The figure was replaced in the revised manuscript.

For Figure B, we have performed RT-PCR with CD11b+-sorted cells with PMA or PMA-Post-PD, two time.  The results of two repeat experiments were similar to each other.  The graph from two repeat experiments is shown figure below, left.

  1. Figure 3C, D. Is it only one n number?

As shown above, we performed the experiments at least three times, which were analyzed statistically and expressed as graphs in Figure 3C and D in the revised manuscript.

  1. Line 287-287, either provide the data for the claim or remove this sentence from the manuscript.

As pointed by the reviewer, we removed the sentence in the revised manuscript.

Discussion:

  1. Overall, the discussion section is very poorly written. This section needs to provide a rich discussion comparing current findings and their importance, as well as citing more literature for comparison and possible mechanisms. The discussion should include a paragraph of limitation of the study.

As pointed by the reviewer, we included the limitation of this study as follows in the revised manuscript:

The limitation of this study might be lack of investigation into the effect of nutrients on THP-1 differentiation and polarization in response to PMA in the presence or absence of PD98059 or Trametinib.  In tumor microenvironment, nutrients, or metabolic environment might be closely related to the MEK-ERK pathways and the efficacy of targeting these pathways.

  1. Line 386, either show the data or remove this claim from the manuscript.

As pointed by the reviewer, we removed the word cytotoxcitiy from the line 386 in the revised manuscript.

  1. Lines 396-398 are confusing, and need to be revised.

As pointed by the reviewer, we improved our expression as follows in the revised manuscript.

Trametinib, the agent targeting MEK1/2 could modulate macrophage populations re-sulting in increased M1 population (Fig. 2F).  Thus, the benefit of targeting MEK1/2 for cancer cells with BRAF mutations and macrophage polarization by MEK1/2 inhibi-tors requires further investigation.

Minor Corrections:

  1. Line 104, 173 correct “5 X 105” Cells

As pointed by the reviewer, we corrected the expression.  Please check this out in the revised manuscript.

  1. Line 119-120 “ No antibodies are Fluorescein conjugated “ in the provided Supplemental table. The same mistake was repeated in the Results section, Line 233-234. If the Author used AI for editing language, please read carefully and disclose the usage of AI in declaration.

As pointed by the reviewer, we corrected the expression and clarified fluorochromes of antibodies against markers in the revised supplementary table and the revised manuscript.  Thank you for your sophisticated guidance for precise expression.  Frankly, we did not get any help from AI program, but editing service from “TEXTCHECK”, an official English editing service company as shown in the certificate below when we submitted our manuscript.

  1. Provide the version of the software used, like Flowjo , TomoStudio X and ImageJ.

As pointed by the reviewer, we updated the software information for this study (FlowJo v9, TomoStudio Xè TumoStudioTM, and ImageJ v1.53, respectively) in detail in the revised manuscript.

  1. Line 137 says PCR products were sequenced. Where is the data for that sequencing? Which platform is used for sequencing?

PCR products were isolated and subjected to sanger sequencing for validation if the product is from the gene that we expected by aligning sequencing results and reference gene.  Some examples were shown in below.

  1. Figure 2E, labeling for PD98059 is incorrect.

As pointed by the reviewer, we corrected the label of Figure 2E in the revised manuscript.  We appreciated your detail guidance.

  1. Line 266, correct Figure 1D to 2D.

As pointed by the reviewer, we corrected that in the revised manuscript.  Check this out in the revised manuscript, please.

I hope that the revised manuscript responds appropriately to the criticisms raised, and am confident that it now demonstrates the PD98059 on PMA-induced polarization to more inflammatory M1 population of macrophages including THP-1 macrophages.  I therefore very much hope that it will be considered suitable for publication in your Journal.

If you have any questions, please do not hesitate to fax me on 82-31-920-2206 or send e-mail.

Thank you for your careful attention to this matter, and I look forward to hearing from you.

Sincerely,

Hye Jin You, Ph.D.

Principal Investigator

Chief of Cancer Microenvironment Branch, Div. of Cancer Biology, Research Institute,

National Cancer Center; Adjunct Professor of Department of Cancer Biomedical Science, National Cancer Center-Graduate School of Cancer Science and Policy (NCC-GCSP), National Cancer Center, 323 Ilsan-ro, Ilsandong-gu, Goyang, Gyeonggi, 10408, South Korea

Tel: 82-31-920-2206

Reviewer 2 Report

Comments and Suggestions for Authors

I am pleased to have been involved in the review of this interesting paper. The manuscript presents results that are contrary to existing reports suggesting that PD98059 stimulation promotes M2 polarization in macrophages. In such cases, it is crucial for the authors to clarify the differences with past studies and to demonstrate the novelty of their findings within the manuscript. Addressing the following concerns will make this manuscript more engaging for a broad readership.

#1 Please provide more detailed explanations of the differences from previous studies. According to another researcher, PD98059 stimulation promotes M2 polarization in macrophages (PMID: 28003382, https://www.ncbi.nlm.nih.gov/pmc/articles/PMC5224968/). This is the opposite of what your results indicate. It is necessary to discuss what might have caused these discrepancies, including the complexity of signaling pathways or the activation of alternative pathways. Additionally, the previous report includes in vivo studies, so please discuss the novelty of your manuscript, which solely comprises in vitro experiments. These aspects could potentially be expanded upon in the discussion section.

#2 In Figure 1D, showing "Adherent cells," both groups (PMA-post-PD and PMA without PD) are indicated to have statistical differences, though the differences appear to be minor. The statistical analyses of several figures in this manuscript could benefit from multi-group evaluations. Please consult a statistical expert to ensure that there are no errors in the statistical processing.

#3 In Figure 2F, the colors and titles in the flow cytometry peaks do not match. For example, what does the highest peak, colored brown, represent?

Comments on the Quality of English Language

NA

Author Response

NATIONAL CANCER CENTER

Cancer Microenvironment Branch, Division of Cancer Biology, Research Institute; Department of Cancer Biomedical Science, NCC-Graduate School of Cancer Science and Policy
323 Ilsan-ro, Ilsandong-gu, Goyang-si, Gyeonggi-do, 10408, Republic of Korea
Phone: +82-31-920-2206, Fax: +82-31-920-2006

May 24th, 2024

Professor, Dr. Edgaras Stankevicius Editor in Chief

Medicina, 

Editorial Office, MDPI Publications.

Dear Professor, Dr. Edgaras Stankevicius Editor in Chief,  

Thank you very much for the valuable comments on our manuscript, medicina-3005169, entitled “THP-1 Monocytic Cells May be Polarized to More Antitumorigenic Macrophages by Serial Treatment With Phorbol-12-Myristate-13-Acetate and PD98059”. 

Based on the reviewer’s comments, we are providing a point-by-point discussion of the points raised from the reviewers.

Reviewer2

I am pleased to have been involved in the review of this interesting paper. The manuscript presents results that are contrary to existing reports suggesting that PD98059 stimulation promotes M2 polarization in macrophages. In such cases, it is crucial for the authors to clarify the differences with past studies and to demonstrate the novelty of their findings within the manuscript. Addressing the following concerns will make this manuscript more engaging for a broad readership.

  1. Please provide more detailed explanations of the differences from previous studies. According to another researcher, PD98059 stimulation promotes M2 polarization in macrophages (PMID: 28003382, https://www.ncbi.nlm.nih.gov/pmc/articles/PMC5224968/). This is the opposite of what your results indicate. It is necessary to discuss what might have caused these discrepancies, including the complexity of signaling pathways or the activation of alternative pathways. Additionally, the previous report includes in vivo studies, so please discuss the novelty of your manuscript, which solely comprises in vitro experiments. These aspects could potentially be expanded upon in the discussion section.

First of all, we are appreciated with the valuable reference.  As pointed by the reviewer, there are some studies to support MEK pathways for M2 polarization promotion while other studies strengthen MEK pathways for proinflammatory response.  Thus we discussed the controversy with two more references including the article suggested by the reviewer in the revised manuscript as shown below.

Interestingly, there are controversies in regards to MEK-ERK pathway on macro-phage polarization.  Manicone and colleagues showed the MEK1/2 inhibitors resulted in greater M2 polarization in alveolar macrophages and bone marrow-derived macro-phages in the presence of IL-4/IL-13 [38], which further supported by another study to show MEK1/2 inhibition on proinflammatory response [39].  In contrast, MEK-ERK driven M2 polarization was selectively blocked by MEK and histone deacetylase inhibitors in THP-1 and macrophages prepared from mouse and human [28].  In this study, we have demonstrated that the MEK1-ERK pathway serves as a target to modulate the M1 population of THP-1 cells under PMA treatment.  The controversy might be caused by a variety of experimental systems and types such as treatment condition and processes for evaluation and so on.

  1. Long, M.E.; Eddy, W.E.; Gong, K.Q.; Lovelace-Macon, L.L.; McMahan, R.S.; Charron, J.; Liles, W.C.; Manicone, A.M. MEK1/2 Inhibition Promotes Macrophage Reparative Properties. J Immunol 2017, 198, 862-872, doi:10.4049/jimmunol.1601059.
  2. De, M.; Serpa, G.; Zuiker, E.; Hisert, K.B.; Liles, W.C.; Manicone, A.M.; Hemann, E.A.; Long, M.E. MEK1/2 inhibition decreases pro-inflammatory responses in macrophages from people with cystic fibrosis and mitigates severity of illness in experimental murine methicillin-resistant Staphylococcus aureus infection. Front Cell Infect Microbiol 2024, 14, 1275940, doi:10.3389/fcimb.2024.1275940.

  1. In Figure 1D, showing "Adherent cells," both groups (PMA-post-PD and PMA without PD) are indicated to have statistical differences, though the differences appear to be minor. The statistical analyses of several figures in this manuscript could benefit from multi-group evaluations. Please consult a statistical expert to ensure that there are no errors in the statistical processing.

As pointed by the reviewer, we have checked our statistical data out precisely and some are improved and the rest of them are kept same in the revised manuscript. 

In Figure 1D (à Figure 1C in the revised manuscript) significance between groups was changed and added as shown in figure left and we replaced this figure in the revised manuscript.  Please check this out in the revised manuscript.  Thank you for valuable comments.   

  1. In Figure 2F, the colors and titles in the flow cytometry peaks do not match. For example, what does the highest peak, colored brown, represent?

As pointed by the reviewer, we clarified data label for better understanding in Figure 2E as shown below and replaced in the revised manuscript.  The highest peak, brown represents cells with PMA treatment.  Please check this improved figure out in the revised manuscript.

I hope that the revised manuscript responds appropriately to the criticisms raised, and am confident that it now demonstrates the PD98059 on PMA-induced polarization to more inflammatory M1 population of macrophages including THP-1 macrophages.  I therefore very much hope that it will be considered suitable for publication in your Journal.

If you have any questions, please do not hesitate to fax me on 82-31-920-2206 or send e-mail.

Thank you for your careful attention to this matter, and I look forward to hearing from you.

Sincerely,

Hye Jin You, Ph.D.

Principal Investigator

Chief of Cancer Microenvironment Branch, Div. of Cancer Biology, Research Institute,

National Cancer Center; Adjunct Professor of Department of Cancer Biomedical Science, National Cancer Center-Graduate School of Cancer Science and Policy (NCC-GCSP), National Cancer Center, 323 Ilsan-ro, Ilsandong-gu, Goyang, Gyeonggi, 10408, South Korea

Tel: 82-31-920-2206

Round 2

Reviewer 1 Report

Comments and Suggestions for Authors

Dear Authors,

The provided explanations for the review questions were mostly satisfactory. However, there are still some minor but critical mistakes that need to be corrected.

1.       Figure 1 has been revised but the legend is not revised appropriately. In the revised figure “there is no dot plot in 1C and there is no any 1D figure”, revise the figure legend 1 appropriately.

2.       What is the logic for Adherent vs Non-Adherent THP-1 cell culture, and why is it important?

As studied by a lot of researchers, THP-1 cells are very useful model for monocytes/macrophages and convenient to deal with.  However, the experimental approaches are descriptive and had better be evaluated for clarity.  In this study, we tested several aspects of THP-1 differentiation and polarization in response to PMA from culture conditions to treatment time, and so on. Clearly, we showed THP-1 needs adherent condition for efficient differentiation and survival in response to PMA by showing cells in 50-ml tubes with less differentiation and poor viability during treatment time (Fig. 1).

Reply to Authors: While the explanation is satisfactory, the purpose of the question was to include this reasoning/argument in your results section. This will help readers better understand your questions.

3.       Figure 2A, there is no clear separation for CD206 high population. How the gating has been decided.  It seems the M2 population is just by error. Provide FMO or other control for deciding the gating line in supplemental.

As suggested by the reviewer, we showed the control as figure below.  THP1 cells were exposed to PMA and harvested for FACS analyses.  Some cells were not stained at all and used as negative control of each antibody signal.  The other cells were stained with fluorochrome-conjugated antibodies against CD11b, CD206, HLA-DRα, and PD-1.  As shown in figure below, each antibody clearly showed antibody-specific fluorochrome signal. The signal from CD206 was relatively weaker than other markers however, clearly existed.  Thus our gating strategy was applied in this study. 

Reply to Authors: Please include the CD206 histogram in the supplemental materials. preferably, provide a dot plot figure showing both negative and positive results. This will help prevent readers from doubts about the correct gating strategy.

4.       I could not understand why Trametinib is used and what the question is for using Trametinib. Describe in detail.

In this study, we have demonstrated the involvement of MEK-ERK pathway on altered populations by PD98059 (specific for MEK1) in PMA-stimulated THP-1 cells in terms of polarized subsets.  Trametinib in PMA-stimulated THP-1 cells, showed increased M1 population in PMA-stimulated THP-1 cells, supporting the idea, MEK/ERK might be important for macrophage polarization to anti-cancer, and proinflammatory subset not only for differentiation.

By showing effect of Trametinib, our results were further confirmed.  In addition, the results from trametinib-experiments, are meaningful to provide the evidence for future direction in the therapeutic strategy against cancer with RTK, Ras, Raf alteration and certain immune microenvironment.  

Reply to Authors: While the explanation is satisfactory, the purpose of the question was to articulate your question briefly before introducing Trametinib in your results, “that why you performed that experiment”.

5.       Figure 3C, D. Is it only one n number?

As shown above, we performed the experiments at least three times, which were analyzed statistically and expressed as graphs in Figure 3C and D in the revised manuscript.

Reply to Authors: While the images shown are satisfactory, provide these images as supplemental or original gel blot deposits for transparency.

6.        Line 137 says PCR products were sequenced. Where is the data for that sequencing? Which platform is used for sequencing?

PCR products were isolated and subjected to Sanger sequencing for validation if the product is from the gene that we expected by aligning sequencing results and reference gene.  Some examples were shown in below.

Reply to Authors: While the explanation and Sanger sequencing images seem good, provide this in the supplemental figure.

7.       The graphical abstract provided is good, provide it either as a “graphical abstract” or conclusion figure, whichever is suitable as per journal editorial policy.

Author Response

May 28th, 2024

Professor, Dr. Edgaras Stankevicius Editor in Chief

Medicina, 

Editorial Office, MDPI Publications.

Dear Professor, Dr. Edgaras Stankevicius Editor in Chief,  

Thank you very much for the valuable comments on our manuscript, medicina-3005169, entitled “THP-1 Monocytic Cells May be Polarized to More Antitumorigenic Macrophages by Serial Treatment With Phorbol-12-Myristate-13-Acetate and PD98059”. 

Based on the reviewer’s comments, we are providing a point-by-point discussion of the points raised from the reviewers.

Reviewer 1:

  1. Figure 1 has been revised but the legend is not revised appropriately. In the revised figure “there is no dot plot in 1C and there is no any 1D figure”, revise the figure legend 1 appropriately.

As pointed out by the review, we revised the figure legend 1 properly in the revised manuscript.  Thank you for the comments.

  1. What is the logic for Adherent vs Non-Adherent THP-1 cell culture, and why is it important?

As studied by a lot of researchers, THP-1 cells are very useful model for monocytes/macrophages and convenient to deal with.  However, the experimental approaches are descriptive and had better be evaluated for clarity.  In this study, we tested several aspects of THP-1 differentiation and polarization in response to PMA from culture conditions to treatment time, and so on. Clearly, we showed THP-1 needs adherent condition for efficient differentiation and survival in response to PMA by showing cells in 50-ml tubes with less differentiation and poor viability during treatment time (Fig. 1).

Reply to Authors: While the explanation is satisfactory, the purpose of the question was to include this reasoning/argument in your results section. This will help readers better understand your questions.

As suggested by the reviewer, we added our reasoning in the introduction as well as results as follows in the revised manuscript.  Please check this out in the results “line 197-200”.

In Lines, 197-2000,

However, the experimental approaches vary in each study.  THP-1 differentiation in response to PMA, had better be evaluated based on several aspects, including treatment time, culture conditions, to ensure reproducibility.

  1. Figure 2A, there is no clear separation for CD206 high population. How the gating has been decided. It seems the M2 population is just by error. Provide FMO or other control for deciding the gating line in supplemental.

As suggested by the reviewer, we showed the control as figure below.  THP1 cells were exposed to PMA and harvested for FACS analyses.  Some cells were not stained at all and used as negative control of each antibody signal.  The other cells were stained with fluorochrome-conjugated antibodies against CD11b, CD206, HLA-DRα, and PD-1.  As shown in figure below, each antibody clearly showed antibody-specific fluorochrome signal. The signal from CD206 was relatively weaker than other markers however, clearly existed.  Thus our gating strategy was applied in this study.

Reply to Authors: Please include the CD206 histogram in the supplemental materials. preferably, provide a dot plot figure showing both negative and positive results. This will help prevent readers from doubts about the correct gating strategy.

As suggested by the review, we improved supplementary figure by providing histograms as well as dot plots to show the specific staining against CD11b, CD206, HLA-DRα, and PD-1 as follows.  We added this as supplementary figure S2 with legend in the revised manuscript.  Thank you for the comments.

  1. I could not understand why Trametinib is used and what the question is for using Trametinib. Describe in detail.

In this study, we have demonstrated the involvement of MEK-ERK pathway on altered populations by PD98059 (specific for MEK1) in PMA-stimulated THP-1 cells in terms of polarized subsets.  Trametinib in PMA-stimulated THP-1 cells, showed increased M1 population in PMA-stimulated THP-1 cells, supporting the idea, MEK/ERK might be important for macrophage polarization to anti-cancer, and proinflammatory subset not only for differentiation.

By showing effect of Trametinib, our results were further confirmed.  In addition, the results from trametinib-experiments, are meaningful to provide the evidence for future direction in the therapeutic strategy against cancer with RTK, Ras, Raf alteration and certain immune microenvironment. 

Reply to Authors: While the explanation is satisfactory, the purpose of the question was to articulate your question briefly before introducing Trametinib in your results, “that why you performed that experiment”.

As suggested by the reviewer, we added description why we consider the drug “Trametinib in this study as shown below in the revised manuscript. 

Results Line 274-276

Next, we considered to get further insight into the clinical application of anticancer drugs on macrophage polarization, by targeting MEK1 pathways.

  1. Figure 3C, D. Is it only one n number?

As shown above, we performed the experiments at least three times, which were analyzed statistically and expressed as graphs in Figure 3C and D in the revised manuscript.

Reply to Authors: While the images shown are satisfactory, provide these images as supplemental or original gel blot deposits for transparency.

As suggested by the review, we added the repeat experimental results as a file as original blot deposits at this resubmission process.

  1. Line 137 says PCR products were sequenced. Where is the data for that sequencing? Which platform is used for sequencing?

PCR products were isolated and subjected to Sanger sequencing for validation if the product is from the gene that we expected by aligning sequencing results and reference gene.  Some examples were shown in below.

Reply to Authors: While the explanation and Sanger sequencing images seem good, provide this in the supplemental figure.

As suggested by the review, we added the Sequencing results as the supplementary figure S3 in the revised manuscript.  Thank you for your comments.

  1. The graphical abstract provided is good, provide it either as a “graphical abstract” or conclusion figure, whichever is suitable as per journal editorial policy.

As suggested by the reviewer, we tried to summarize our finding as a graphical abstract as shown below.

I hope that the revised manuscript responds appropriately to the criticisms raised, and am confident that it now demonstrates the PD98059 on PMA-induced polarization to more inflammatory M1 population of macrophages including THP-1 macrophages.  I therefore very much hope that it will be considered suitable for publication in your Journal.

If you have any questions, please do not hesitate to fax me on 82-31-920-2206 or send e-mail.

Thank you for your careful attention to this matter, and I look forward to hearing from you.

Sincerely,

Hye Jin You, Ph.D.

Principal Investigator

Chief of Cancer Microenvironment Branch, Div. of Cancer Biology, Research Institute,

National Cancer Center; Adjunct Professor of Department of Cancer Biomedical Science, National Cancer Center-Graduate School of Cancer Science and Policy (NCC-GCSP), National Cancer Center, 323 Ilsan-ro, Ilsandong-gu, Goyang, Gyeonggi, 10408, South Korea

Tel: 82-31-920-2206
